# A method for evaluating breast cancer screening strategies using screen-preventable loss of life

Kimbroe J. Carter[1,2,3]*, Frank Castro[3], Roy N. Morcos[4,5‡]

1 Department of Pathology, Northeast Ohio Medical University, Rootstown, Ohio, United States of America, 2 School of Technology, Kent State University Trumbull Campus, Warren, Ohio, United States of America, 3 Medical Decision Making Society of Youngstown Ohio, Saint Elizabeth Youngstown Hospital, Youngstown, Ohio, United States of America, 4 Department of Family and Community Medicine, Northeast Ohio Medical University, Rootstown, Ohio, United States of America, 5 Saint Elizabeth Boardman Family Medicine Residency Program, Youngstown, Ohio, United States of America

☯ These authors contributed equally to this work.
‡ RNM also contributed equally to this work.
* mdmyo2@hotmail.com

**Data Availability Statement:** Data is provided in Supplemental S1 File

**Funding:** Author KJC, Grant M2018-3 Mercy Health Youngstown, LLC Medical Research Committee. Author KJC, Mercy Health Foundation

## Abstract

The objective of this study is to describe how screen-preventable loss of life (screen-PLL) can be used to analyze the distribution of life savings with mammographic screening. The determination of screen-PLL with mammography is possible using a natural history model of breast cancer that simulates clinical and pathologic events of this disease. This investigation uses a Monte Carlo Markov model with data from the Surveillance, Epidemiology, and End Results Program; American Cancer Society; and National Vital Statistics System. Populations of one million women per screening strategy are simulated over a lifetime with mammographic screening based on current guidelines of the American Cancer Society (ACS), United States Preventive Services Task Force (USPSTF), triennial screening from age 50–70, and no screening. Screen-PLL curves are generated and show guideline performance over a lifetime. The screen-PLL curve with no screening is determined by tumor discovery through clinical awareness and has the highest values of screen-PLL. The ACS and USPSTF strategies demonstrate screen-PLL curves favoring the elderly. The curve for triennial screening is more uniform than the ACS or USPSTF curves but could be improved by adding screen(s) at either end of the 50–70 age range. This study introduces the use of screen-PLL as a tool to improve the understanding of screening guidelines and allowing a more balanced allocation of life savings across an aging population. The method presented shows how screen-PLL can be used to analyze and potentially improve breast cancer screening guidelines.

## Introduction

The goal of screening is to detect early disease when treatment is more likely to be beneficial or lifesaving [1, 2]. The opportunity to save life-years through screening is expressed

Mahoning Valley, https://foundation.mercy.com/youngstown.aspx, The funders had no role in study design, data collection and analysis, decision to publish, or preparation of the manuscript.

**Competing interests:** The authors have declared that no competing interests exist.

mathematically as screen-preventable loss of life (screen-PLL). If an effective intervention is available, the difference in years between individuals' premature deaths and the underlying population's life expectancy is a measure of the potential life savings.

This idea of calculating preventable loss of life for a disease is not new. Examples of this principle are found in analyzing the changing mortality of tuberculosis in the 1940s and more recently in measuring the burden of disease, such as chronic obstructive pulmonary disease [3, 4]. However, application of this concept to mammography screening and breast cancer survival has not been widely described in the medical literature.

Since the introduction of mammography screening in the 1960s, clinical trials have shown mortality reduction, with a few studies showing conflicting results [5–7]. The debate about the use and starting age of mammography screening continues to the present day, as there remains no agreed standard to measure screening performance over a lifetime [8–11]. The United States Preventive Services Task Force (USPSTF) recommends biennial screening for women aged 50 to 74 with average-risk women aged 40–49 encouraged to discuss the advisability of screening with their providers [7]. The American Cancer Society (ACS) recommends annual screening from age 45 to 55, followed by biennial mammography until life expectancy is less than 10 years [12]. In addition, the ACS allows for the opportunity to start screening with annual mammography between the ages of 40 and 44 and to continue yearly mammography at age 55 and older. A strategy of triennial screening from ages 50–70 is similar to the recommendation of The National Health Service in the United Kingdom [13].

In reality, data needed to calculate screen-PLL for breast cancer such as the size and lethality of undetected tumors, time of metastasis, and expected ages of death from breast cancer or alternative causes are not clinically available. However, a computer model of breast cancer's natural history can evaluate these factors and permits insights not possible in clinical practice. A published model of breast cancer screening [14] is adapted to measure screen-PLL.

The objective of this study is to describe a computer modeling approach to determine the screen-preventable life savings achievable within an aging population of women screened by the ACS and USPSTF guidelines, as well as triennial screening and no screening.

## Methods

Screen-PLL is the number of preventable months of life lost to breast cancer from progressive tumors that are mammographically detectable at an early stage and curable. In this study, the screen-PLL value for a simulated individual is calculated at all ages. An assumption is that mammographic detectability starts at a tumor size of 0.2 cm [15]. However, some undetected tumors may lead to distant metastases and premature death from breast cancer. For each woman who dies prematurely, screen-PLL is calculated as the difference between her expected age of death from general mortality and her age of death from breast cancer, as seen in Fig 1. In this figure, screen-PLL is zero until the tumor reaches 0.2 cm. Subsequently, screen-PLL increases to the difference between her expected non-breast cancer death age and her breast cancer death age and remains constant. Screen-PLL returns to zero with metastasis or detection by mammography or clinical signs and symptoms. Detection eliminates the need for further screening. In the model, metastatic disease is assumed incurable since death in most breast cancer cases is associated with metastases [16–18]. Once metastases occur, further screening offers no life-saving benefit. With detection or with presence of metastatic disease, the opportunity to prevent a breast cancer death through screening ends.

As an example, consider a 60-year-old woman who has a localized progressive primary tumor that is 0.2 cm in diameter. Without screening, metastatic disease is predicted by the model to occur at age 65 and death from breast cancer at age 72. This woman's expected age of

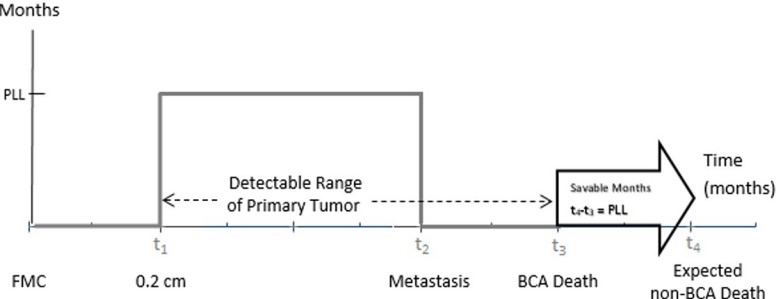

**Fig 1. Screen-preventable loss of life in relation to disease progression.** FMC, first malignant cell; BCA, breast cancer; PLL, preventable loss of life. Mammographic detectability begins at age $t_1$ with a tumor size of 0.2 cm. The detectable range of a primary tumor is from 0.2 cm ($t_1$) to BCA death ($t_3$). Screen-PLL is the age difference between the expected non-BCA death and BCA death, defined as $t_4$-$t_3$. Screen-PLL is 0 from the FMC to the tumor size of 0.2 cm, the value of $t_4$-$t_3$ from 0.2 cm to detection or metastasis, and 0 thereafter.

death from other causes in the model is 92. This is a case of premature death from breast cancer that could have been prevented by mammography. The screen-PLL is 240 months or 20 years, calculated as (92–72) ×12. The screen-PLL of 240 months exists for this individual starting with detectability at age 60 and extending until metastasis at age 65. Once metastases occur, screening mammography is considered of no value in preventing breast cancer death, and screen-PLL is zero thereafter.

A cohort's screen-PLL at a given age is the sum of all individuals' screen-PLLs. The cohort's screen-PLL curve is constructed by plotting the screen-PLL over time. Screen-PLL curves are created for three groups of one million women each, following the ACS, USPSTF and triennial strategies. The ACS guideline is modeled according to the base recommendation and does not include the options for earlier screening and for continued annual mammography at age 55 and older. A screen-PLL curve is also created for an unscreened cohort of one million women in whom clinical signs and symptoms are the only means of detection.

The characteristics of the ACS, USPSTF, and triennial screen-PLL curves are analyzed. A flat curve indicates that screen-PLL is distributed evenly across ages, resulting in an equitable allotment of life savings to the population.

The study uses a published Monte Carlo Markov natural history model for breast cancer, calibrated using statistics from the Surveillance, Epidemiology, and End Results Program (SEER), the American Cancer Society and the National Vital Statistics System [14, 19–21]. In the model, a woman begins at birth with no cancer [14]. Breast cancer arises with a first malignant cell and follows Gompertzian growth with disease progression dependent on tumor volume [22–24]. Age-specific incidence data for breast cancer among average-risk white women in the United States are used to determine the age of onset of clinical disease [25]. The proportion of non-progressive tumors, which are non-lethal and contribute to overdiagnosis, is assumed in our model to be 30%, based on a finding in an analysis of SEER data [26]. It is assumed that during the study, women are 100 percent compliant with their assigned screening strategy. Sensitivity of mammography increases with tumor diameter, starting at 0.2 cm, and is assumed higher for post-menopausal women [14]. Stopping ages for the ACS guideline rule, 10-year or less life-expectancy, were calculated using life table data [14]. Tumor detection results from either mammography screening or from clinical signs and symptoms. The disease is classified into one of seven clinical stages using the American Joint Committee on Cancer's criteria [27]. Death may occur from either metastatic breast cancer or from other causes. In the case of a breast cancer death, life years lost to breast cancer are calculated as the difference between the projected death age from general mortality and the death age from breast cancer.

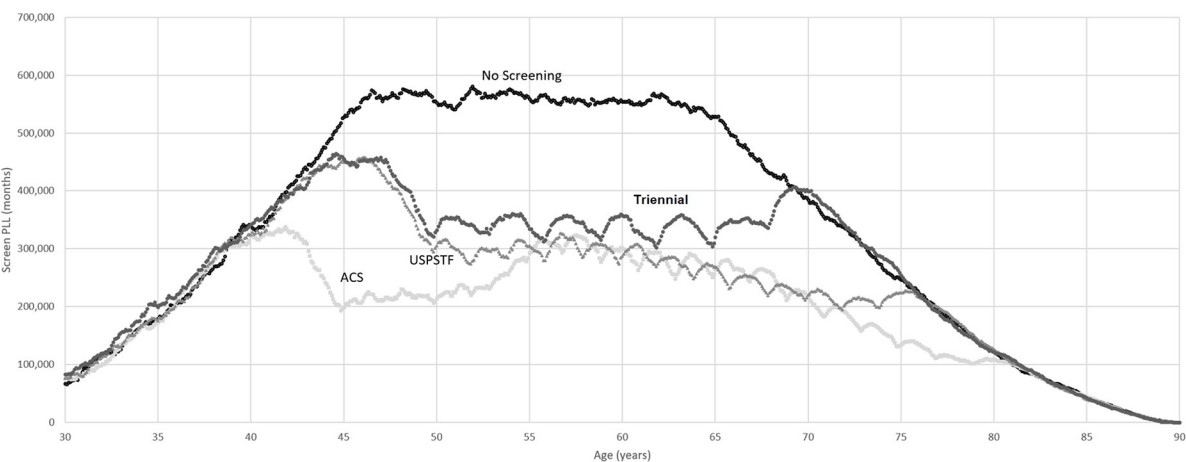

**Fig 2. Breast cancer screen-PLL for the ACS screening guideline, USPSTF screening guideline, triennial screening strategy, and no screening.** One million women in each group. Screen-PLL, screen-preventable loss of life; ACS, American Cancer Society; USPSTF, United States Preventive Services Task Force.

## Results

Fig 2 shows the distributions of screen-PLL in months for cohorts of one million women each, following the ACS, USPSTF, and triennial strategies, as well as for an unscreened cohort of one million women labeled as No Screening. At birth, there is no screen-PLL. As the cohort ages and breast cancer incidence rises, the screen-PLL increases. For the unscreened cohort, the screen-PLL at age 30 is approximately 75,000 life-months. The screen-PLL continues to increase until a plateau of about 575,000 life-months from approximately age 45 to 65. After age 65, screen-PLL rapidly decreases and approaches zero. The USPSTF curve follows the No Screening curve until about age 43. After that age, the USPSTF curve decreases before the onset of screening. Shortly after screening stops at age 74, the USPSTF and No Screening curves converge. The ACS curve decreases significantly with the start of annual screening at age 45 and converges with the USPSTF curve with the start of biennial screening at age 55. The curves stay in relative alignment until about age 74, when USPSTF screening ends. The ACS curve subsequently dips and eventually converges again with the USPSTF curve and the No Screening curve at around age 80 when ACS screening ends. The triennial strategy PLL curve follows the USPSTF curve closely until screening begins at age 50. The triennial PLL curve remains slightly above the USPSTF curve throughout and converges with No Screening at age 70.

## Discussion

Screening mammography remains an important preventive strategy despite some enduring controversies. Since a mammography guideline determines the number of possible screening tests, improving uniformity in the distribution of life savings across age groups ensures equitability. This notion has attracted little attention in the design of screening strategies, as if all participants would benefit equally simply by participating in a screening program. This is clearly not the case. Guideline recommendations determine the distribution of life savings from screening within a population. The concept of screen-PLL highlights the differences between guidelines.

The ACS and USPSTF guidelines continue screening for women into their 70s when, due to limited life expectancy, the screen-PLL is lower than at any other age. This favors elderly

women. Screen-PLL for ACS, USPSTF, and triennial strategies are highest for women in their 40s. This non-uniformity in screen-PLL indicates disproportionate allocation of screening benefits. Age-related deviations in uniformity of screen-PLL indicate opportunities to improve equitability of screening benefits. Screening is best implemented in an age range of higher screen-PLL. One might conclude that for a given number of screens and with all other factors held constant, the guideline with the most uniform screen-PLL is the most efficient.

Conceptually, factors that determine screen-PLL include breast cancer incidence and mortality, general life expectancy, clinical symptoms, screen timing, and the sensitivity of mammography. In the model, mammographic resolution is assumed to be 0.2 cm and determines the magnitude of screen-PLL. Improved resolution would increase screen-PLL, while decreased resolution would lower it. However, mammographic resolution does not alter the shape of the screen-PLL curve, but simply shifts the screen-PLL curves up or down.

Screen-PLL accumulates with the onset and growth of cancers. This is due to the natural history of the disease and is not dependent upon the initiation of screening. Although the potential to reduce screen-PLL exists outside the age boundaries of screening guidelines, it is limited by the available screening technologies, the relatively small number of cases in young women, and the reduced life-expectancy in older women.

In an unscreened cohort, screen-PLL increases as age-specific breast cancer incidence increases, and plateaus from about age 45 to 65. This plateau is due to tumor discovery through clinical manifestations along with the offsetting effects of increasing tumor incidence and diminishing life expectancy. After age 65, shortened life expectancy from general mortality causes a rapid decrease in screen-PLL. Accelerating this trend is the fact that fewer elderly women with screen-preventable disease will die of their cancer.

In the screen-PLL curves for ACS, USPSTF, and triennial screening, there is a reduction in screen-PLL each time mammography is performed and an increase during the interim between screens, resulting in the peaks and valleys seen in Fig 2. Each time a screen is performed, tumors are detected, changing the screen-PLL for these individuals to zero and reducing the screen-PLL for the cohort.

There are limitations to this study. First, only women in the United States with average breast cancer risk are modeled. Thus, the results may vary for other populations. Additionally, as with any model, the results are dependent upon the underlying assumptions and model calibration. The existence of non-progressive tumors and occurrence of overdiagnosis are unpredictable for an individual. Estimates of the overdiagnosis level in a population vary greatly [26, 28–33]. In this study, an overdiagnosis rate of 30% is assumed. Regardless of the actual overdiagnosis value, only women at risk of death from breast cancer influence screen-PLL. By definition, women overdiagnosed by screening do not die from breast cancer and thus do not contribute to preventable life years lost to breast cancer. As a result, screen-PLL curves for a given screening strategy with different overdiagnosis rates generate superimposable screen-PLL curves as observed in validation testing.

In this study we have examined screen-PLL as a method for assessing breast cancer screening guidelines within the framework of a natural history model. Crucially, the study simply explores concepts and consequences of screen-PLL in delivering one benefit, preventable loss of life, to a population of aging women. Contrasts of screen-PLL curves among three major guidelines illustrate the principle of equity of health care delivery among young and aging women. Equity across ages in screening guidelines is not directly considered in guideline development, but we suggest that it should be. Perhaps this gap exists because a clinical decision metric such as screen-PLL has not been developed.

This study of screen-PLL has not considered monetary factors for a screening program and provides no cost-effectiveness implications. While screen-PLL makes inequities in a breast

cancer screening strategy obvious, it does not evaluate the economics of screening strategies. However, the concept of screen-PLL could be applied in the context of cost-effectiveness in future studies. Such an analysis would need to consider the cost of therapies at various stages of disease as well as the costs and harms of screening. Additionally, the concept of screen-PLL may be used in future guideline development to offer additional life savings for the same number of screens.

## Conclusions

This study broadens the concept of screen-PLL as a method to improve the understanding of breast cancer screening guidelines in distributing life savings across an aging population. By analyzing screen-PLL curves, the health benefits of mammography screening may be more equitably allocated.

## Supporting information

**S1 File. Screen-PLL curve data.** Excel.xlsx workbook with monthly data used to create screen-PLL curves for Fig 2.
(XLSX)

## Acknowledgments

The authors would like to thank Dr. Timothy Barreiro, Associate Professor of Internal Medicine, Northeast Ohio Medical University and Dr. Michael Kavic, Professor Emeritus of Surgery, Northeast Ohio Medical University for final review of the manuscript. We would also like to thank Drs. David Gemmel and Vincent Vanek, Department of Medical Research, St. Elizabeth Hospital Youngstown for continued support.

## Author Contributions

**Conceptualization:** Kimbroe J. Carter, Frank Castro, Roy N. Morcos.

**Data curation:** Frank Castro.

**Formal analysis:** Frank Castro.

**Funding acquisition:** Kimbroe J. Carter.

**Methodology:** Kimbroe J. Carter, Frank Castro.

**Project administration:** Kimbroe J. Carter.

**Resources:** Kimbroe J. Carter.

**Software:** Kimbroe J. Carter.

**Supervision:** Kimbroe J. Carter.

**Validation:** Kimbroe J. Carter, Frank Castro, Roy N. Morcos.

**Writing – original draft:** Kimbroe J. Carter, Frank Castro, Roy N. Morcos.

**Writing – review & editing:** Kimbroe J. Carter, Frank Castro, Roy N. Morcos.

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
