## [Decision Letter · Decision Letter 0]

15 Oct 2020

PONE-D-20-25853

A Method for Evaluating Breast Cancer Screening Strategies Using Screen-Preventable Loss of Life

PLOS ONE

Dear Dr. Carter,

Thank you for submitting your manuscript to PLOS ONE. After careful consideration, we feel that it has merit but does not fully meet PLOS ONE’s publication criteria as it currently stands. Therefore, we invite you to submit a revised version of the manuscript that addresses the points raised during the review process.

We look forward to receiving your revised manuscript.

Kind regards,

Eugenio Paci, MD

Academic Editor

PLOS ONE

Additional Editor Comments:

I found the proposal of Screen-Preventable Loss of Life estimate very interesting and the tool potentially useful in order to assess the impact of different screening guidelines. The interesting example of USPSTF and ACS is a significant example. However, as the reviewer discussed, each guideline has many aspects to be considered and areas of uncertainties which must be considered.

I guess that the new version of the paper should answer to some the suggestions , because it is important if the new tool might be considered in the future comparison of screening guidelines. In my view, the authors should assess the impact of the evaluation not only in terms of gain in term of better impact. The question is relevant when two guidelines which are different in minimal aspects in terms of benefit are justified? A guideline could increase minimally the benefit but with increasing costs or harms or , which is also important, practical complexity for the woman and service.

Journal Requirements:

Reviewers' comments:

Reviewer's Responses to Questions

**Comments to the Author**

1. Is the manuscript technically sound, and do the data support the conclusions?

Reviewer #1: Partly

2. Has the statistical analysis been performed appropriately and rigorously? 

Reviewer #1: No

3. Have the authors made all data underlying the findings in their manuscript fully available?

Reviewer #1: No

4. Is the manuscript presented in an intelligible fashion and written in standard English?

Reviewer #1: Yes

5. Review Comments to the Author

Reviewer #1: PONE-D-20-25853, A Method for Evaluating Breast Cancer Screening Strategies Using Screen-Preventable Loss of Life

Line 72: The authors should completely describe the American Cancer Society guideline, which also encourages women to make an informed decision about beginning screening at ages 40-44 (but should begin no later than 45), while also having the choice to continue annual screening after age 55, with the same stopping criteria. I appreciate that you can’t practically model that. In the methods section you might note that guidelines are modeled according to the base recommendation and do not accommodate shared decision periods.

Line 89: By this definition of screen-PLL, wouldn’t a breast cancer death at any age be a premature death? Loss of life is the projected remaining years of life among the cohort that is still living and has not died from breast cancer, but is it estimated from the expected date of death if the cancer had not been detected by screening? The example is clear in the context of a 0.2 cm tumor (very small), but the methods should describe how the model works for other occult tumor sizes that are not metastatic, unless all screen detected cancers have these parameters. Consider the ACS guideline, which could be judged to be a little extreme with respect to 10 + year longevity as a criteria for quitting--any estimate of Screen-PLL < 10 years would, by the guideline definition, should not have invoked a referral to screening, and any longevity over 85 is off the table for the USPSTF. Personally, avoiding a premature death with some meaningful number of years less than 10 (allowing for the treatment and recovery period) would likely be judged to be worth it by the patient, but also obviously, we have numerous studies in the literature of women with severe life-limiting comorbidity, who should not have been referred to screening, who die from another cause within a 1-2 years of their diagnosis. So, the question, is there a threshold for a screen-PLL that is not worth pursuing because it may lead to more harm than benefit? Also, I’m guessing that the role of therapy is assumed to be stage-specific and fixed, so that the contribution of screening, and earlier detection, is fixed. That is worth mentioning if that is the case.

Line 92: The way screen-PLL is expressed could use a bit more explanation, I think. The authors state that screen-PLL increases to the difference between her expected non-breast cancer death age and her (expected?) breast cancer death age. In your example, the expected breast cancer death age is set at 12 years if she is detected with a 0.2 cm tumor. But, in this example, it seems that screen-PLL can only decrease as a function of age at diagnosis, specifically younger age at diagnosis, and to a lesser extent, tumor size (with screen-PLL also decreasing with < tumor size) within the detectable range, which runs up to metastasis. Does this model assume all symptomatic breast cancers are metastatic? Is metastatic here synonymous with distant disease? The logic of the PLL being a function of T4-T3 is obvious, but how T3 is estimated could be made clearer.

Line 124: A description of the basis for comparison between the guidelines seems to be missing. Does the model assume that the age to begin and end screening results in 100 percent adherence with screening and 100% screen detection of cancers? If so, please explain, and explain how stopping ages were modeled for the ACS guideline. Is it just modeled on a life-table?

Line 133: Is it reasonable to state that the PLL beings to accumulate as women increase in age, even though guidelines do not endorse screening for average risk women until 40? The authors might acknowledge that the true potential of reducing PLL is within the boundaries of the recommended screening protocols, before and after which we have to accept some breast cancer deaths and life years lost are unavoidable.

Line 177: The first limitation could be judged to be perfunctory and is not worth mentioning since any study done in a single country faces the same global limitation in true generalizability, although you could note that this is a function of the screening recommendations and the burden of disease. I’d explain why it is a limitation, and perhaps just note that the results will vary by population risk and the screening protocol. In some respects, the US is perhaps the largest heterogeneous population in the world. Second, population estimates of overdiagnosis don’t vary greatly because of population differences, they vary greatly due to 1) the limitations of the data to estimate overdiagnosis, 2) the wide range of methodologies applied, which have been judged to range from quite good to quite flawed. Here the authors have chosen a quite flawed estimate (30%), which is many times higher than the estimates from studies that have shown more sophisticated awareness of factors associated with differences in incidence over time in a population exposed to screening vs. not exposed to screening. re are several limitations that the authors have not identified. If the authors have used overdiagnosis rates estimated from SEER because they’re using incidence and mortality from SEER, the model does not require this. I suggest that the authors consider the work of Danish investigators on this issue….. (1) Lynge, E., Beau, A. B., Christiansen, P., von Euler-Chelpin, M., Kroman, N., Njor, S., & Vejborg, I. (2017). Overdiagnosis in breast cancer screening: The impact of study design and calculations. Eur J Cancer, 80, 26-29. doi:10.1016/j.ejca.2017.04.018; (2) Lynge, E., Beau, A. B., von Euler-Chelpin, M., Napolitano, G., Njor, S., Olsen, A. H., . . . Vejborg, I. (2020). Breast cancer mortality and overdiagnosis after implementation of population-based screening in Denmark. Breast Cancer Res Treat. doi:10.1007/s10549-020-05896-9; and (3) Njor, S. H., Paci, E., & Rebolj, M. (2018). As you like it: How the same data can support manifold views of overdiagnosis in breast cancer screening. Int J Cancer, 143(6), 1287-1294. doi:10.1002/ijc.31420. The authors could do a sensitivity analysis (0, 5, 10, 19, and 30)…these estimates are all in the literature.

There are some limitations the authors have not mentioned. At line 120, the authors state, “Age

specific incidence data for breast cancer among average-risk white women in the United States

are used to determine the age of onset of clinical disease.” The authors should acknowledge that only half, and maybe not that many breast cancers are screen detected, or preferred, detected among women attending screening in any give year (screen detected and interval cancers). So, the SEER data are a mix of high risk and average risk women, and a mix of screen-detected and clinically detected disease. This, and the implications for the model, should be acknowledged.

Lines 183-189: I don’t really see these as study limitations. Perhaps shift to another part of the discussion, or move to the methodology.

6. PLOS authors have the option to publish the peer review history of their article (what does this mean?). If published, this will include your full peer review and any attached files.

Reviewer #1: No

---

## [Author Response · Author response to Decision Letter 0]

1 Nov 2020

November 1, 2020

Authors’ responses to reviewer and editor comments:

Reviewer #1: PONE-D-20-25853, A Method for Evaluating Breast Cancer Screening Strategies Using Screen-Preventable Loss of Life

1. The authors should completely describe the American Cancer Society guideline, which also encourages women to make an informed decision about beginning screening at ages 40-44 (but should begin no later than 45), while also having the choice to continue annual screening after age 55, with the same stopping criteria. I appreciate that you can’t practically model that. In the methods section you might note that guidelines are modeled according to the base recommendation and do not accommodate shared decision periods.

Authors’ Response: We have added these details to the description of the ACS guideline in the Introduction. We have noted in the Methods that our model considers only the base recommendation.

2. By this definition of screen-PLL, wouldn’t a breast cancer death at any age be a premature death? 

Authors’ Response: Yes, death from breast cancer is considered premature in our model, as stated in the Introduction. No additional changes to the manuscript have been made regarding this comment.

3. Loss of life is the projected remaining years of life among the cohort that is still living and has not died from breast cancer, but is it estimated from the expected date of death if the cancer had not been detected by screening? 

Authors’ Response: Screen-PLL is evaluated for all women as the number of preventable months of life lost to breast cancer from progressive tumors that are mammographically detectable at an early stage and curable. It is calculated as the difference between her expected age of death from general mortality and her age of death from breast cancer. However, once the cancer is detected by screening or clinical symptoms, screen-PLL is zero, as further screening provides no additional benefit. The term preventable has been added to the Methods in the definition of screen-PLL for clarity. 

4. The example is clear in the context of a 0.2 cm tumor (very small), but the methods should describe how the model works for other occult tumor sizes that are not metastatic, unless all screen detected cancers have these parameters. 

Authors’ Response: Occult tumors less than 0.2 cm diameter are assumed in the model to be undetectable by mammography, therefore are not considered screen-preventable. Occult tumors greater than 0.2 cm but not metastatic are precisely the tumors that contribute to screen-PLL. No changes have been made to the manuscript regarding this comment.

 

5. Consider the ACS guideline, which could be judged to be a little extreme with respect to 10 + year longevity as a criteria for quitting--any estimate of Screen-PLL < 10 years would, by the guideline definition, should not have invoked a referral to screening, and any longevity over 85 is off the table for the USPSTF. Personally, avoiding a premature death with some meaningful number of years less than 10 (allowing for the treatment and recovery period) would likely be judged to be worth it by the patient, but also obviously, we have numerous studies in the literature of women with severe life-limiting comorbidity, who should not have been referred to screening, who die from another cause within a 1-2 years of their diagnosis. So, the question, is there a threshold for a screen-PLL that is not worth pursuing because it may lead to more harm than benefit? 

Authors’ Response: The intent of the study was to compare screen-PLL curves for major screening guidelines. Developing an ideal strategy with respect to optimal start and stop ages as well as screening frequency is beyond the scope of this study. The concept of screen-PLL may be beneficial in helping develop future breast cancer screening guidelines. The judgment of what amount of screen-PLL is worth pursuing lies in the domain of cost-effectiveness. The Discussion was expanded to include some of these concepts. 

6. Also, I’m guessing that the role of therapy is assumed to be stage-specific and fixed, so that the contribution of screening, and earlier detection, is fixed. That is worth mentioning if that is the case.

Authors’ Response: The role of stage-specific therapy is not considered. The model assumption is that metastatic disease is the cause of breast cancer death and that non-metastatic cancers are curable. No changes have been made to the manuscript regarding this comment.

 

7. The way screen-PLL is expressed could use a bit more explanation, I think. The authors state that screen-PLL increases to the difference between her expected non-breast cancer death age and her (expected?) breast cancer death age. In your example, the expected breast cancer death age is set at 12 years if she is detected with a 0.2 cm tumor. But, in this example, it seems that screen-PLL can only decrease as a function of age at diagnosis, specifically younger age at diagnosis, and to a lesser extent, tumor size (with screen-PLL also decreasing with < tumor size) within the detectable range, which runs up to metastasis. Does this model assume all symptomatic breast cancers are metastatic? Is metastatic here synonymous with distant disease? The logic of the PLL being a function of T4-T3 is obvious, but how T3 is estimated could be made clearer.

Authors’ Response: The model does not assume that all symptomatic breast cancers are metastatic. As discussed in the methods section, tumors may be detected by screening or with clinical symptoms. We define metastatic disease as distant disease, which arises in the model when the woman’s primary tumor reaches pre-determined volume thresholds. This has been clarified in Methods. The age of breast cancer death in the model (t3) for a given woman is based on her metastatic disease. Death occurs when her metastatic tumor load reaches a terminal level, which was explained in Methods. No further additions were made. 

We appreciate the reviewer’s comments that screen-PLL decreases with advancing age as life-expectancy shortens and that younger women would tend to have a higher screen-PLL, given longer life-expectancies. These concepts were presented in Discussion and no additional explanations were made. 

Screen-PLL is constant during the detectable tumor range and does not vary with primary tumor size. Screen-PLL is zero once metastatic disease occurs. The rationale behind screen-PLL has been expanded throughout the revised manuscript.

8. A description of the basis for comparison between the guidelines seems to be missing. Does the model assume that the age to begin and end screening results in 100 percent adherence with screening and 100% screen detection of cancers? If so, please explain, and explain how stopping ages were modeled for the ACS guideline. Is it just modeled on a life-table?

Authors’ Response: The following was added to Methods to address these items of guideline adherence, screen detection, and ACS stopping age modeling:

It is assumed that during the study, women are 100 percent compliant with their assigned screening strategy. Sensitivity of mammography increases with tumor diameter, starting at 0.2 cm, and is assumed higher for post-menopausal women. Stopping ages for the ACS guideline rule, 10-year or less life-expectancy, were calculated using life table data. 

 

9. Is it reasonable to state that the PLL beings to accumulate as women increase in age, even though guidelines do not endorse screening for average risk women until 40? The authors might acknowledge that the true potential of reducing PLL is within the boundaries of the recommended screening protocols, before and after which we have to accept some breast cancer deaths and life years lost are unavoidable.

Authors’ Response: PLL does begin to accumulate even before the age at which screening is recommended based on the guidelines. The potential to reduce PLL exists outside the age boundaries of screening guidelines, however it is limited by the available screening technologies, the relatively small number of cases in young women, and the limited life-expectancy in older women. These notions have been added to the Discussion.

10. The first limitation could be judged to be perfunctory and is not worth mentioning since any study done in a single country faces the same global limitation in true generalizability, although you could note that this is a function of the screening recommendations and the burden of disease. I’d explain why it is a limitation, and perhaps just note that the results will vary by population risk and the screening protocol. In some respects, the US is perhaps the largest heterogeneous population in the world. 

Authors’ Response: We agree with the reviewer that this is a relatively minor limitation which is common to similar studies performed within a single country. We have modified the text to reflect that results may vary with different populations. 

11. Second, population estimates of overdiagnosis don’t vary greatly because of population differences, they vary greatly due to 1) the limitations of the data to estimate overdiagnosis, 2) the wide range of methodologies applied, which have been judged to range from quite good to quite flawed. Here the authors have chosen a quite flawed estimate (30%), which is many times higher than the estimates from studies that have shown more sophisticated awareness of factors associated with differences in incidence over time in a population exposed to screening vs. not exposed to screening. re are several limitations that the authors have not identified. If the authors have used overdiagnosis rates estimated from SEER because they’re using incidence and mortality from SEER, the model does not require this. I suggest that the authors consider the work of Danish investigators on this issue….. (1) Lynge, E., Beau, A. B., Christiansen, P., von Euler-Chelpin, M., Kroman, N., Njor, S., & Vejborg, I. (2017). Overdiagnosis in breast cancer screening: The impact of study design and calculations. Eur J Cancer, 80, 26-29. doi:10.1016/j.ejca.2017.04.018; (2) Lynge, E., Beau, A. B., von Euler-Chelpin, M., Napolitano, G., Njor, S., Olsen, A. H., . . . Vejborg, I. (2020). Breast cancer mortality and overdiagnosis after implementation of population-based screening in Denmark. Breast Cancer Res Treat. doi:10.1007/s10549-020-05896-9; and (3) Njor, S. H., Paci, E., & Rebolj, M. (2018). As you like it: How the same data can support manifold views of overdiagnosis in breast cancer screening. Int J Cancer, 143(6), 1287-1294. doi:10.1002/ijc.31420. The authors could do a sensitivity analysis (0, 5, 10, 19, and 30)…these estimates are all in the literature.

Authors’ Response: The authors are aware of the reviewer’s concerns regarding the vagaries of breast cancer overdiagnosis. As noted, the model was standardized on SEER incidence and mortality data. A published study for overdiagnosis of SEER data estimated overdiagnosis at 30 percent. 

Importantly, only women at risk of death from breast cancer influence screen-PLL. By the very definition of overdiagnosis, women overdiagnosed by screening do not die from breast cancer and thus do not contribute to preventable life years lost to breast cancer. As a result, screen-PLL curves for a given screening strategy with different overdiagnosis rates generate superimposable screen-PLL curves. This was confirmed with internal validation testing.

The concept of breast cancer overdiagnosis is interesting and contentious. Many reasons contribute to the inconsistencies of overdiagnosis observations/measurements as observed in the articles referenced by the reviewer. The dynamics of overdiagnosis are not the intent of this study, and overdiagnosis does not influence the characteristics of screen-PLL. 

The Discussion was modified to include these concepts.

12. There are some limitations the authors have not mentioned. At line 120, the authors state, “Age

specific incidence data for breast cancer among average-risk white women in the United States

are used to determine the age of onset of clinical disease.” The authors should acknowledge that only half, and maybe not that many breast cancers are screen detected, or preferred, detected among women attending screening in any give year (screen detected and interval cancers). So, the SEER data are a mix of high risk and average risk women, and a mix of screen-detected and clinically detected disease. This, and the implications for the model, should be acknowledged.

Authors’ Response: We agree that incidence data reflects a mix of screen detected and interval cancers. The population that we have modeled includes both groups of women. No changes have been made to the manuscript regarding this comment.

13. I don’t really see these as study limitations. Perhaps shift to another part of the discussion, or move to the methodology.

Authors’ Response: This concept has been moved out of limitations and to another part of the discussion.

 

Additional Editor Comments:

1. I guess that the new version of the paper should answer to some the suggestions, because it is important if the new tool might be considered in the future comparison of screening guidelines. In my view, the authors should assess the impact of the evaluation not only in terms of gain in term of better impact. The question is relevant when two guidelines which are different in minimal aspects in terms of benefit are justified? A guideline could increase minimally the benefit but with increasing costs or harms or, which is also important, practical complexity for the woman and service.

Authors’ Response: The authors have examined screen-PLL as a method for assessing breast cancer screening guidelines within the framework of a natural history model. Crucially, the study simply explores concepts and consequences of screen-PLL in delivering one outcome, preventable loss of life, to a population of aging women. Contrasts of screen-PLL curves among three major guidelines illustrate the notion of equity of health care delivery among young and aging women. Equity across ages in screening guidelines is not a notion considered directly in guideline development, but we suggest that it should be. Perhaps this gap exists because a clinical decision tool handling these screen-PLL concepts is not available. 

This study of screen-PLL has not considered cost factors for a screening program such as the cost of screening and subsequent therapy and provides no cost-effectiveness implications. While screen-PLL makes inequities in a breast cancer screening strategy obvious, it does not evaluate the economics of breast cancer screening strategies. The judgment of what amount of screen-PLL is worth pursuing lies in the domain of cost-effectiveness. The notion of screen-PLL could be applied in the context of cost-effectiveness in future studies. A cost-effectiveness screen-PLL analysis would need to consider the cost of therapies at various stages of disease as well as the costs and harms of screening. 

The Discussion was modified to include these concepts.

Authors’ Response: Relevant data is submitted with the manuscript revision.

---

## [Decision Letter · Decision Letter 1]

16 Nov 2020

A method for evaluating breast cancer screening strategies using screen-preventable loss of life

PONE-D-20-25853R1

Dear Dr. Carter,

We’re pleased to inform you that your manuscript has been judged scientifically suitable for publication and will be formally accepted for publication once it meets all outstanding technical requirements.

Kind regards,

Eugenio Paci, MD

Academic Editor

PLOS ONE

Additional Editor Comments (optional):

This is a very interesting and important tool in order to compare different screening policies. 

Reviewers' comments:

Reviewer's Responses to Questions

**Comments to the Author**

1. If the authors have adequately addressed your comments raised in a previous round of review and you feel that this manuscript is now acceptable for publication, you may indicate that here to bypass the “Comments to the Author” section, enter your conflict of interest statement in the “Confidential to Editor” section, and submit your "Accept" recommendation.

Reviewer #1: All comments have been addressed

2. Is the manuscript technically sound, and do the data support the conclusions?

Reviewer #1: Yes

3. Has the statistical analysis been performed appropriately and rigorously? 

Reviewer #1: Yes

4. Have the authors made all data underlying the findings in their manuscript fully available?

Reviewer #1: Yes

5. Is the manuscript presented in an intelligible fashion and written in standard English?

Reviewer #1: Yes

6. Review Comments to the Author

Reviewer #1: (No Response)

7. PLOS authors have the option to publish the peer review history of their article (what does this mean?). If published, this will include your full peer review and any attached files.

Reviewer #1: **Yes: **Robert A. Smith

---

## [Editor Report · Acceptance letter]

20 Nov 2020

PONE-D-20-25853R1 

A method for evaluating breast cancer screening strategies using screen-preventable loss of life 

Dear Dr. Carter:

I'm pleased to inform you that your manuscript has been deemed suitable for publication in PLOS ONE. Congratulations! Your manuscript is now with our production department. 

Kind regards, 

on behalf of

Dr. Eugenio Paci 

Academic Editor

PLOS ONE